# Molecular and Morphological Characterization of Introgression Lines with Resistance to Bacterial Leaf Blight and Blast in Rice

**DOI:** 10.3390/plants12163012

**Published:** 2023-08-21

**Authors:** Yasaswini Vishnu Priya Varanasi, Subhakara Rao Isetty, Padmashree Revadi, Divya Balakrishnan, Shaik Hajira, Madamsetty Srinivasa Prasad, Gouri Shankar Laha, Puvvada Perraju, Uma Maheshwar Singh, Vikas Kumar Singh, Arvind Kumar, Raman Meenakshi Sundaram, Jyothi Badri

**Affiliations:** 1ICAR—Indian Institute of Rice Research (ICAR-IIRR), Hyderabad 500030, India; 2Jawaharlal Nehru Krishi Vishwa Vidyalaya, Rewa 486001, India; 3International Rice Research Institute South Asia Regional Centre (ISARC), Varanasi 221006, India; 4International Rice Research Institute South Asia Hub (IRRISAH), Hyderabad 221106, India; v.k.singh@irri.org; 5International Crop Research Institute for Semi-Arid Tropics (ICRISAT), Hyderabad 502324, India

**Keywords:** multi parent selective inter-crossing, near isogenic lines (NILs), marker assisted forward breeding, yield, principal component analysis, SNPs, background selection

## Abstract

The present study evaluates marker assisted forward breeding (MAFB)-derived disease resistant introgression lines (ILs) which do not have the targeted resistance genes for bacterial blight (*xa5 + xa13 + Xa21*) and blast (*Pi2 + Pi9* + *Pi54*). The ILs were derived in the background of two elite rice cultivars, Krishna Hamsa [Recurrent Parent 1 (RP1)] and WGL 14 (RP2), involving multi-parent inter-crossing. Molecular characterization with gene specific markers for seven reported resistance genes each for bacterial blight (*Xa33*, *Xa38*, *xa23*, *Xa4*, *xa8*, *Xa27* and *Xa41*) and blast (*Pi1*, *Pi20*, *Pi38*, *Pib*, *Pitp*, *Pizt* and *Pi40*) revealed the presence of *xa8* and *Xa38*, in addition to the targeted *xa5*, *xa13* and *Xa21* for bacterial blight resistance and *Pi1*, *Pi38*, *Pi40*, *Pi20*, *Pib* and *Pipt*, in addition to the targeted *Pi9* and *Pi54*, for blast resistance in various combinations. A maximum of nine resistance genes *xa5 + Xa21 + Pi54 + xa8 + Pipt + Pi38* + *Pi1 + Pi20* + *Pib* was observed in RP1-IL 19030 followed by eight genes *xa5 + xa13 + Xa21 + xa8 + Pi9 + Pipt + Pi1 + Pi20* in two RP2-ILs, 19344 and 19347. ANOVA revealed the presence of significant variability for all the yield traits except “days to 50% flowering” (DFF). Box plots depicted the seasonal differences in the phenotypic expression of the yield traits. There was significant positive association of grain yield with days to flowering, tiller number and panicle number. Thousand grain weight is also significantly and positively correlated with grain yield. On the contrary, grain yield showed a significantly negative association with plant height. Multi-parent selective inter-crossing in the present study not only led to the development of high yielding disease resistant ILs but also enhanced recovery of the recurrent parent via selection for essential morphological features. More than 90.0% genetic similarity in the ILs based on SNP-based background selection demonstrated the success of multi-parent selective intercrossing in the development of disease resistant NILs.

## 1. Introduction

Rice (*Oryza sativa* L.) is a global staple cereal for over half the world’s population. Rice production needs to be increased by 42% from the current level by 2050 to feed the ever-increasing population [1]. Rice production is highly sensitive to many biotic and abiotic factors. Yield losses due to various biotic stresses are estimated as being up to 52%, of which nearly 31% is due to diseases such as bacterial blight, blast, sheath blight and tungro disease [2]. Changing climatic conditions are contributing to the emergence of new virulent diseases. Blast caused by the fungus *Magnaporthe oryzae* expresses as spindle-shaped spots with a grey/white central part and brownish borders. This fungus infects almost all the parts of rice plant at almost all stages of growth, from seedling to maturity [3], and accounts for 70–80% of all yield losses. Bacterial blight caused by *Xanthomonas oryzae* pv. *oryzae* affects the rice plant at all stages of growth, with maximum damage at the tillering stage. In the initial stages, it is noticeable as water-soaked streaks that spread from the leaf tips and margins, becoming larger, and eventually releasing a milky ooze that dries into yellow droplets. Characteristic grayish-white lesions then appear on the leaves, signaling the late stages of infection, where leaves dry out and die. Yield losses in severely infected fields range from 20 to 30%, but can reach as high as 80% [4].

Breeding rice cultivars with disease resistance possessing good agronomic yield potential has been the need of the hour for decades [5]. In this context, genetic improvement is the most effective method, one that enables a reduction in the adverse effect on the environment through the reduced usage of pesticides which directly or indirectly affect our ecology and ecosystem [6]. However, with this approach, durability of host resistance and resurgence of the pathogen are the major limitations, often resulting in a breakdown of the resistance within a few years. Therefore, resistance breeding requires continuous efforts to evaluate the diversity within the germplasm [7]. Understanding the genetics of disease mechanisms and stacking of broad-spectrum disease resistance genes could lead to faster development of rice varieties with multiple disease resistance. About 45 bacterial-blight resistance genes have been discovered in rice [2], and some of them have been successfully introgressed into the genomes of commercial rice cultivars. To date, more than 100 blast resistance genes have been identified, and 27 have been cloned and characterized [8].

Marker-assisted selection (MAS) has been successfully employed in the recent past, both for sequential [9,10] and simultaneous introgression [6,11,12,13] of resistance/tolerance to two or more biotic or abiotic stresses. Simultaneous introgression of multiple traits involves repeated cycles of selective inter-crossing to combine traits from different donors, genotyping with foreground markers to track alleles and stringent phenotyping across different generations [11]. Marker-assisted forward breeding (MAFB) for simultaneous multiple trait introgression involving four to six donors has been reported for bacterial blight, blast and gallmidge in Swarna [6]; blast, gallmidge and drought in Naveen [12]; drought, bacterial blight and blast in Lalat [13]; and bacterial blight, blast and drought resistance/tolerance in Krishna Hamsa [11]. An interesting observation in these reports discusses resistance to bacterial blight and/or blast in some of the introgression lines (ILs) despite the absence of targeted resistance genes.

The success of any marker-assisted breeding depends not only on the selection of lines with target trait introgression, but also on the selection of such lines with desirable agro-morphological traits culminating in the development of cultivars. This requires multi-season field evaluation in replicated trials. Since grain yield is a complex agronomic trait, as determined by the ultimate expression of its individual component traits, various selection criteria should be employed in the identification of promising genotypes with a desirable combination of traits. Phenotypic selection strategies based on essential morphological traits and grain quality traits in recovering the elite parent phenotype have largely been successful [13,14,15].

Further, trait introgression into an elite cultivar background typically requires multiple generations of backcrossing, leading to the development of near isogenic lines (NILs) with morphological similarity in important agronomic traits and maximum genetic similarity. Breeding by selective introgression was earlier proposed [16] for simultaneous introgression of two or more complex traits, a process which uses an improved version of backcross breeding. Contrarily, ILs developed through selective inter-crossing involving multiple parents showed multi-trait introgressions possessing more than 95% genetic similarity with recurrent parent [11]. Recently, a simulated breeding pipeline demonstrated the success of intercrossing in trait introgression using fewer resources compared to the traditional backcrossing strategies [17]. Keeping in view the aforesaid, the aims of the present study are (1) to determine genes associated with resistance to bacterial blight and blast diseases in the ILs; (2) to select ILs with high yield and desirable plant phenotype; and (3) to assess the genetic similarity between ILs (having maximum phenotype recovered for essential morphological features) and their respective recurrent parents.

## 2. Material and Methods

### 2.1. Materials Used in the Study

The experimental material included ILs with resistance to bacterial blight and/or blast, developed using MAFB in the background of two elite cultivars, ‘Krishna Hamsa’ and ‘WGL 14′. The development of the ILs in the background of ‘Krishna Hamsa’ involving six donors was described in our previous work [11]. The six donors included IRBB60 with *xa5*, *xa13* and *Xa21* for bacterial blight; Tetep with *Pi9* and *Pi54* for blast resistance; IR 71033-121-15-B with *Bph20* and *Bph21* for BPH resistance; IR 96321-1447-561-B-1 with *qDTY1.1* and *qDTY3.1*; IR 81896-96-B-B-195 with *qDTY2.1*; and IR 74371-46-1-1-13 with *qDTY12.1* for the incorporation of drought tolerance. In case of ‘WGL 14’, which is a high-yielding variety with desirable cooking quality traits and medium slender grains, nine donors were used in MAFB. The nine donors viz., Improved Samba Mahsuri (ISM) with *xa5*, *xa13* and *Xa21* for bacterial blight; Tetep with *Pi9* and *Pi54*; RP Patho-1 with *Pi2* for blast; Raathu Heenathi with *Bph3* and *Bph17* for brown plant hopper; RP 5924-24 with *Gm4*; RP 5924-25 with *Gm8* for gallmidge resistance; IR 96321-1447-561-B-1 with *qDTY1.1* and *qDTY3.1*; IR 81896-96-B-B-195 with *qDTY2.1* and IR 74371-46-1-1-13 with *qDTY12.1* for yield under drought stress were used in the selective intercrossing, with WGL 14 as recurrent parent. The breeding strategy was described earlier in detail in our previous work [11,18]; in brief, it involved crossing of all the donors to a common elite parent, foreground selection, phenotypic selection for target traits and repeated cycles of inter-crossing of the F_1_s between them.

Bacterial blight screening was done both under field conditions and glass house conditions using the artificial clip inoculation method, along with recurrent parents, Krishna Hamsa and WGL 14, and checks/controls Improved Samba Mahsuri (resistant check/control) and TN1 (susceptible check/control), as described in our previous work [11]. A highly virulent, local isolate of BB pathogen *Xanthomonas oryzae* pv. *oryzae* (*Xoo*) IX-020, maintained at ICAR-IIRR, was used for screening the ILs. Under field conditions, leaf tips of three plants in each IL were cut with scissors dipped in a BB suspension of 10^9^ cfu/mL at 40 days after transplanting (DAT), coinciding with the maximum tillering stage. Similarly, under glass house conditions, 45–50-day-old seedlings were inoculated. Blast screening was performed in a universal blast nursery (UBN) facility. In raised nursery beds with a row spacing of 10 cm, one row of the susceptible check (HR-12) was planted between every four entries, and also along the borders, to facilitate the build-up of inoculum for uniform and rapid spread of the disease. *Magnaporthe oryzae* isolate ‘IIRR-31’ is a highly virulent isolate collected from major blast disease hot spots and maintained at ICAR-IIRR. The inoculum, with a concentration of 1 × 10^5^ spore/mL, was sprayed onto young seedlings at four leaf stages using a fine sprayer, and high relative humidity was maintained for disease development. Tetep was used as a resistance check.

### 2.2. Molecular Characterization

In the present investigation, the 91 ILs were essentially derived using MAFB. At the same time, some of the ILs without the targeted gene introgressions, viz., *xa5*, *xa13* and *Xa21* for bacterial resistance, and *Pi2*, *Pi9* and *Pi54* for blast resistance, were also selected based on their phenotypic resistance. To ascertain the cause of resistance in such ILs, they were profiled with other gene-specific markers for seven reported resistance genes each for bacterial blight (*Xa33*, *Xa38*, *xa23*, *Xa4*, *xa8*, *Xa27* and *Xa41*) and blast (*Pi1*, *Pi20*, *Pi38*, *Pib*, *Pitp*, *Pizt* and *Pi40*) (Appendix A). Leaf DNA extraction, DNA quantification and PCR amplification were performed as described earlier [11].

### 2.3. Morphological Characterization

The present investigation also focused on the evaluation of a subset of ILs for yield traits to select high-yielding ILs. From among the 91 ILs, 41 with desirable agronomic traits and phenotypic acceptability were selected for yield evaluation, along with two recurrent parents, ‘Krishna Hamsa’ (RP1) and ‘WGL 14′ (RP2). Additionally, three elite cultivars, including ISM (check 1), Samba Mahsuri (check 2) and Swarna (check 3) were used as yield checks. The experiment was conducted during wet season (WS) 2020 and dry season (DS) 2021 at ICAR–IIRR. In WS, sowing and planting were conducted on 8 July 2020 and 17 August 2020, respectively. In DS, sowing and planting were conducted on 2 January 2021 and 8 February 2021, respectively. Experiments were carried out in a randomized block design (RBD) with two replications and 20 hills per row, maintaining 20 cm × 15 cm spacing. Appropriate agronomical operations and timely plant protection measures were followed to raise the healthy crop.

Agro-morphological data was collected as to days to fifty-percent flowering (DFF), as the number of days from the date of sowing to complete exsertion of the panicle in 50% of the total number of plants in the net plot; plant height (PH) in cm was measured at the time of plant maturity by using a meter scale, and measuring from the base of the plant to the tip of the panicle; panicle length (PL) was measured in cm at the time of plant maturity from the base of the panicle to the tip of the last spikelet before harvesting; tiller number (TN) was the total number of tillers for each group of four randomly selected plants at the end of the active tillering stage; the total number of panicles per plant (PN) was counted for each randomly selected group of four plants at the time of harvesting; grain number (GN) was the total number of filled grains per panicle for each group of randomly selected four panicles at maturity; thousand grain weight (TW) was measured in g for four replicates of one thousand well-filled grains selected at random from each entry after drying; and grain yield or plot yield (PY) was measured in g/m^2^ as yield of the total plants grown in one square-meter plot. Data was subjected to various statistical analyses using PB tools (Version 1.4, http://bbi.irri.org/products, accessed on 20 April 2023). Analysis of variance (ANOVA) was computed with the level of significance tested at 5% and 1% using the F test to assess the significant differences among the ILs and other genetic variability parameters, viz., mean, range, heritability, etc., as estimated. Correlation analysis was performed to determine association among the yield component traits, and principal component analysis (PCA) was conducted to identify the positive and negative contributions of the component traits to the diversity of ILs.

### 2.4. Background Selection

ILs having comparable yield and other agro-morphological traits of their respective recurrent parents were assessed for recurrent parent genome recovery (RPG) using SNPs. Next-generation sequencing (NGS) platform Illumina NovaSeq was used for genotyping by sequencing (GBS). The sequenced raw data was processed to obtain high-quality clean reads using Trimmomatic v0.38. The reads of the samples were aligned to the reference sequence using BWA MEM (version 0.7.17), with minimum seed length set to 32 and shorter split hits marked as secondary (parameters: −k 32 −M). The mpileup utility of Samtools (v 0.1.18) was used to identify SNPs from the sorted BAM file of the sample. The SNPs were filtered based on a minimum read depth of 5 and a quality threshold of 25. Recurrent parent genome recovery for each IL was calculated as the percent of number of SNPs in the IL equal to the recurrent parent at the corresponding loci to that of the total number of SNPs.

## 3. Results

Of the 91 ILs in the present investigation, 48 ILs were in the background of ‘Krishna Hamsa’ [recurrent parent 1 (RP 1)] and 43 ILs were in the background of ‘WGL 14′ [recurrent parent 2 (RP 2)]. The screening results of the RP1-ILs were described in detail in our earlier work [9]. Among the 48 RP1-ILs, nine are resistant to bacterial blight and blast, 13 to bacterial blight and 26 to blast. Among the 43 RP2-ILs, 19 are resistant to bacterial blight and blast, 14 to bacterial blight and ten to blast (Figure 1). Thus, a total of 28 bacterial blight and blast resistant ILs, 27 bacterial blight resistant ILs and 36 blast resistant ILs from both the recurrent parents were subjected to molecular characterization with targeted bacterial blight resistance (*xa5*, *xa13* and *Xa21*) and blast resistance (*Pi54*, *Pi2* and *Pi9*), as well as reported bacterial blight resistance (*Xa33*, *Xa38*, *xa23*, *Xa4*, *xa8*, *Xa27* and *Xa41*) and other blast resistance (*Pi1*, *Pi20*, *Pi38*, *Pib*, *Pitp*, *Pizt* and *Pi40*) genes.

### 3.1. Molecular Characterization

In all the ILs subjected to molecular characterization with gene-specific markers of 20 genes, there was no amplification of three bacterial blight resistance genes (*xa23*, *Xa4* and *Xa27*), or two blast resistance genes (*Pi2* and *Pizt*), and polymorphism was not observed in two bacterial blight resistance genes, viz., *Xa33* and *Xa41*. The presence of the remaining 15 genes in various combinations is described in the three categories of (1) ILs with resistance to both bacterial leaf blight and blast diseases, (2) ILs with resistance to bacterial leaf blight disease and (3) ILs with resistance to blast disease.

#### 3.1.1. ILs with Resistance to Both Bacterial Leaf Blight and Blast Diseases

The profiling of the bacterial blight and blast resistant ILs with gene specific markers revealed an absence of targeted bacterial blight and blast resistance genes in one RP1-IL 19246, instead, it was marker-positive to *Xa38 + Pipt* + *Pi1*. Similarly, in two RP2-ILs, 19103 and 19160, there was no introgression of any of the targeted BB-R genes. In six RP1-ILs (19007, 19019, 19020, 19025, 19031 and 19378) and five RP2-ILs (19070, 19072, 19101, 19104 and 19483), the targeted blast resistance gene introgression was not observed. On the other hand, they did possess the favorable alleles of *Pi1*, *Pi20*, *Pipt*, *Pi40*, *Pib* and *Pi38* in various combinations. In the remaining ILs, targeted and other resistance genes of bacterial blight and blast were observed in various combinations, and the presence of the greatest number (nine) of resistance genes, *xa5 + Xa21 + Pi54 + xa8 + Pipt* + *Pi38* + *Pi1 + Pi20* + *Pib*, was observed in RP1-IL 19030, followed by eight resistance genes, *xa5 + xa1 3 + Xa21 + Pi9 + xa8 + Pipt* + *Pi1 + Pi20*, in two RP2-ILs, 19344 and 19347 (Table 1).

#### 3.1.2. ILs with Resistance to Bacterial Leaf Blight Disease

The targeted bacterial blight resistance genes were not introgressed in two RP1-ILs, 19233 and 19247. IL 19233 was found to have *Xa8 + xa8*, and IL 19247 was found to have *xa8*. In the remaining ILs, targeted and other bacterial blight resistance genes were observed in various combinations, and the greatest number (four) of resistance genes, *xa5 + xa13 + Xa21 + xa8*, was present in two RP2-ILs, 19379 and 19346, in addition to *Xa21 + xa5 + xa8 + Xa38* in RP2-IL 19075 (Table 2).

#### 3.1.3. ILs with Resistance to Blast Disease

There were ten RP1-ILs (19018, 19021, 19023, 19024, 19026, 19180, 19181, 19211, 19396 and 19411) and five RP2 ILs (19058, 19059, 19060, 19068 and 19490) with zero introgression of targeted blast resistance genes, but possessing the favorable alleles of *Pi1*, *Pi20*, *Pipt*, *Pi40*, *Pib* and *Pi38* in various combinations. In the remaining ILs, targeted and other resistance genes of blast were observed in various combinations, and the greatest number (six) of resistance gene combinations, *Pi54 + Pipt + Pi1 + Pib + Pi20 + Pi38*, was found in the RP2-IL19128 (Table 3).

### 3.2. Morphological Characterization

The objective of morphological characterization was to identify disease resistant high-yielding ILs and NILs which can be released as cultivars. Thus, a subset of 41 resistant ILs with phenotypic superiority for desirable agronomic traits were evaluated for agro-morphological traits along with recurrent parents and yield checks during WS 2020 and DS 2021.

#### 3.2.1. Analysis of Variance (ANOVA) and Correlation Analysis

Season-wise ANOVA revealed significant differences among the genotypes for all the traits except for DFF, while significant differences were found for all traits in the pooled analysis (Table 4). Coefficient of variation varied from 6.79 (DFF) to 26.63% (GN) in the wet season (WS) of 2020 (Appendix A), 7.65 (PL) to 28.17% (PY) in the dry season (DS) of 2021 (Appendix A), and 8.55 (PH) to 30.46% (PY) in the pooled analysis (Appendix A). PY had a significant positive correlation with both tiller number (TN) and panicle number (PN) in WS, and only with PN in DS. In the pooled analysis, PY had a significant positive correlation with DFF, TN, PN and TW, and significant negative correlation with PH. Of note, DFF showed significantly positive correlations with all traits, except for plot yield, in the DS. The significantly positive correlations were also observed with most traits, but not for panicle number, thousand grain weight and plot yield in the WS. The same trend was also observed in the pooled analysis, except that DFF showed no significant associations with PH and PL, while a significantly negative association was noted between DFF and TW (Table 5).

#### 3.2.2. Principal Component Analysis (PCA)

The PCA extracted eight principal components (PCs), equal to the number of studied traits in both the seasons-based and the pooled analyses. The first PC (PC1) captured a maximum variation of 31.53, 31.91 and 34.33% in WS, DS and pooled analysis, respectively, and the first three major PCs (PC1, PC2 and PC3) accounted for a cumulative variance of 65.43, 67.54 and 71.2% in WS, DS and pooled analysis, respectively (Table 6). The positive effect of the variables DFF, PH, PL and GN, and the negative effect of the variables TN, PN, TW and PY for PC1 were commonly observed in both the seasons-based as well as the pooled analysis (Figure 2A).

Biplots also revealed the distribution of the ILs based on respective RP, with, however, some overlaps. In WS, two RP2-ILs, viz., 19206 and 19248, grouped together with RP1-ILs, while seven RP1-ILs (19420, 19347, 19072, 19411, 19030, 19182 and 19024) clustered alongside RP2-ILs. Except ISM, the remaining two checks were quite distinct from the rest of the genotypes. In DS, three RP2-ILs (19283, 19483 and 19344) grouped together with RP1-ILs, and two RP1-ILs (19072 and 19347) clustered alongside RP2-ILs. Swarna (check 3) was quite distinct from rest of the genotypes (Figure 2B).

#### 3.2.3. Performance of the Introgression Lines for Yield Traits

Seasonal differences in the performance of the ILs for yield traits were highly significant. Box plots for DFF (Figure 3A) revealed a medium duration, with a range of 89–119 and mean of 103 ± 7.00 in WS (Appendix A), while late flowering with a range of 100–135 days and a mean of 117 ± 10.12 was observed in the DS (Appendix A). In the pooled analysis, it varied from 98 (RP1-IL 19013) to 127 days (RP2-IL 19347), with a mean of 110 ± 11.17 (Appendix A). In comparison with their respective RPs, all of the RP1-ILs recorded similar duration as that of Krishna Hamsa, while two ILs (19182 and 19461) showed late maturity. In case of RP2-ILs, six ILs recorded a DFF of mid-early to medium duration, while WGL 14 is a late maturing cultivar, and the duration of the 13 ILs matched with it.

Dwarf to semi-dwarf stature was observed in WS, with a range of 77–113 cm and mean of 90 ± 7.11 cm (Appendix A). However, in DS, an overall reduction in PH was observed, with an overall variation from 71–103 cm and mean of 85 ± 7.15 cm (Appendix A) (Figure 3B). In the pooled analysis, measurements varied from 76 (RP2-IL19283) to 108 cm (RP2-IL 19353), with a mean of 88 ± 7.48 cm (Appendix A). Among RP1-ILs, six (19180, 19181, 19408, 19420, 19461 and 19030) were taller than, and the remaining 16 were similar to, their RP. In case of RP2-ILs, four ILs (19128, 19162, 19283 and 19344) were shorter than RP, two ILs (19353 and 19483) were taller than RP, and the remaining 13 ILs were similar to RP.

PL was comparable in both of the seasons (Figure 3C). It varied from 16.45 to 32.5 cm, with a mean of 23 ± 2.56 cm in the wet season (Appendix A), and from 18.03 to 27.18 cm, with a mean of 22 ± 1.7 cm, in the dry season (Appendix A). In the pooled analysis, it varied from 17.71 (RP2-IL 19283) to 27.21 cm (RP2-IL 19345), with a mean of 22 ± 2.17 cm (Appendix A). In case of RP1-ILs, eight ILs (19023, 19026, 19180, 19181, 19182, 19408, 19420 and 19461) recorded longer panicles than their RP, and the remaining 14 ILs had similar PL to that of their RP. In case of RP2-ILs, eight ILs (19284, 19346, 19144, 19283, 19429, 19447, 19483 and 19487) recorded smaller PL than RP, two (19345 and 19072) a longer PL than RP and there were no significant differences in the remaining nine ILs (19484, 19485, 19128, 19353, 19103, 19162, 19344, 19347 and 19448) as compared to the RP.

An increase in TN in DS was observed when compared to WS (Figure 3D). Outliers in TN were observed during both the seasons, with a wide range. The ranges were 6–21 and 12–26, with means of 12 ± 1.94 (Appendix A) and 16 ± 1.39 (Appendix A) in WS and DS, respectively. In the pooled analysis, it varied from 11 (RP1-19247 and RP2-ILs 19345 and 19162) to 20 (RP1-19025), with a mean of 14 ± 2.4 (Appendix A). In case of RP2-ILs, three ILs (19022, 19020 and 19025) had higher TN, while the remaining values were similar to those of their RP. In case of RP2-ILs, seven ILs recorded higher TN and there were no significant differences in the remaining 12 ILs, as compared to the RP2.

Box plots revealed a wide range in the values for PN during both of the seasons, with outliers only in WS (Figure 3E). PN varied from 6 to 26, with a mean of 12 ± 2.79 in the wet season (Appendix A), while it was between 12 and 21, with a mean of 16 ± 1.72 (Appendix A) in the dry season. In the pooled analysis, it varied from 10 (RP2- ILs: 19162, 19347 and 19128, and RP1-IL: 19247) to 23 (RP1-IL: 19020), with a mean of 14 ± 2.87 (Appendix A). In RP1-ILs, two ILs (19020 and 19025) recorded higher, and the remaining 20 ILs similar, PNs as that of Krishna Hamsa. In RP2-ILs, 7 ILs recorded higher PLs, and the remaining 12 ILs recorded PLs similar to that of WGL14.

Low to medium range in GN was observed with outliers both in the WS and DS seasons (Figure 3F). It varied from 53 to 241, with a mean of 111 ± 29.62 (Appendix A), and from 58 to 182, with a mean of 114 ± 23.47 (Appendix A) in the wet and dry seasons, respectively. In the pooled analysis, GN varied from 70 (RP1-19206) to 169 (RP2-19072), with a mean of 112 ± 26.6 (Appendix A). Four RP1-ILs (19182, 19408, 19461 and 19030) recorded higher, and the remaining 17 ILs recorded similar, GN compared to that of Krishna Hamsa. In RP2-ILs, two ILs (19128 and 19072) recorded higher GN, while five ILs (19284, 9103, 19283, 19447 and 19448) recorded smaller values, and there were no significant differences in the remaining 12 ILs as compared to WGL14.

A wide range in TW was observed (Figure 3G), which varied from 11.9 to 29.45 g, with a mean of 21 ± 4.59 g in the wet season (Appendix A), and from 11.51 to 29.05 g, with a mean of 18 ± 4.01 g in the dry season (Appendix A). In the pooled analysis, TW varied from 11.71 g (RP2-19346) to 27.14 g (RP1-19408), with a mean of 20 ± 4.54 g (Appendix A). With the exception of 19408, the remaining RP1-ILs recorded TWs similar to that of Krishna Hamsa. Among the RP2-ILs, eight ILs recorded higher values, and no significant differences were observed in the remaining 11, compared to WGL 14.

Significant differences in PY were observed (Figure 3H), which ranged from 250 to 756 g/m^2^, with a mean of 439 ± 85.87 g/m^2^ in the wet season (Appendix A), and from 410 to 1081 g/m^2^, with a mean of 609 ± 171.41 g/m^2^, in the dry season (Appendix A). In the pooled analysis, PY varied from 238 (RP2-IL 19447) to 771 g/m^2^ (RP1-IL 19022), with a mean of 524 ± 159.52 g/m^2^ (Appendix A). Seven RP1-ILs recorded PYs greater than, and the remaining 15 ILs recorded PYs similar to, that of Krishna Hamsa. Among the RP2-ILs, 14 ILs recorded PYs greater than, and the remaining five ILs recorded PYs similar to, that of WGL 14.

### 3.3. Background Recovery

In the identification of near-isogenic lines (NILs), agro-morphological similarity in terms of plant type similar to that of the recurrent parent for plant height, days to fifty-percent flowering, and grain type are the four most important attributes, along with genetic similarity, which is explained by genome recovery % relative to the recurrent parent. Agro-morphological characterization revealed similarity for plant type and grain type of six RP1-ILs (19026, 19185, 19206, 19211, 19396 and 19019) and two RP2-ILs (19345 and 19484) with their respective recurrent parents. The six RP1-ILs possessed long slender (LS) grains similar to ‘Krishna Hamsa’ and the two RP2-ILs possessed medium slender (MS) grains similar to ‘WGL 14′, and both demonstrated an on-par or higher yield compared to the respective recurrent parents. Based on 35,330 single nucleotide polymorphism (SNPs), RP genome recovery in the aforementioned ILs varied from 90.70 to 93.26%, indicating the genetic similarity of these ILs with their respective recurrent parents (Table 7).

## 4. Discussion

### 4.1. Molecular Characterization

Marker assisted selection basically implies foreground selection for targeted genes/QTLs; however, in the present study, in addition to the selection of plants based on foreground markers, we also selected some of the plants with missing alleles of the targeted resistance genes for bacterial blight and/or blast. The selection of such plants was based on their resistance reaction to blast disease in UBN and to bacterial blight in field and glass house screening. In earlier studies, resistances to bacterial blight and blast were observed in the ILs of Swarna [6], Lalat [13] and Naveen [12], in which the resistant alleles of the targeted genes were missing. The cause of resistance in such ILs with missing resistant alleles was attributed either to the background effect or to positive epistatic interactions in the genome [11,12,13,16,19,20,21], but the genes associated with resistance were not studied. In another study, genotypic specificity led to the non-recovery of some of the gene/QTL combinations [22]. In the present investigation, we attempted to unravel the cause of resistance in such ILs, since multiple donors were actually utilized in their development.

In the present investigation, there were five bacterial blight-resistant ILs (19246, 19103, 19160, 19233 and 19247) in which there was introgression of the targeted genes *xa5*, *xa13* and *Xa21*. Screening with gene-specific markers of reported bacterial-blight resistance genes revealed the presence of *xa8* in ILs 19103, 19160 and 19247; *Xa38* in IL 19246; and a combination of *xa8 + Xa38* in 19233. In other ILs possessing targeted genes, though not in all of them, the enhanced resistance in the ILs to bacterial blight could be due to the presence of the targeted *xa5*, *xa13* and *Xa21*, as well as two additional resistance genes, *xa8* and *Xa38*, in various combinations, suggesting the predominance of these alleles. Previously, bacterial blight resistance gene profiling across the rice germplasm was mostly confined to only four genes, viz., *Xa4*, *xa5*, *xa13* and *Xa21* [4,23,24,25,26,27,28,29,30,31,32]. However, *Xa1* is the most frequently selected gene, followed by *xa7* > *Xa4* > *Xa10* > *Xa11*, while an increase in the frequency of *xa8* in the released varieties has also been observed [33]. To date, combinations of *xa5 + xa13 + Xa21* or *xa5 + Xa21*; *Xa4 + xa5 + xa13 + Xa21*; and *Xa33 + xa5 + xa13 + Xa21* have been successfully introgressed in several of the commercial cultivars for bacterial blight resistance, and such combinations have been providing durable broad-spectrum resistance to virulent *Xoo* isolates [1,34]. The findings of the present study suggest that *xa8* and *Xa38* can be effectively utilized, either singly or in combination with *xa5*, *xa13* and *Xa21*, in the development of bacterial-leaf-blight-resistant cultivars.

As many as 20 ILs, including five bacterial blight and blast resistant ILs (Table 1) and 15 blast resistant ILs (Table 3), were detected as missing the allele of the targeted blast resistance genes in the present study. Molecular screening with other reported blast resistance genes revealed the presence of six resistance genes, viz., *Pi1*, *Pi38*, *Pi40*, *Pi20*, *Pib* and *Pipt* in various combinations, and could be responsible for their enhanced blast resistance. Consistent with the findings of the present investigation, there are several reports on the prevalence of a number of blast resistance genes having significant associations with blast resistance across a wide array of rice germplasm [35,36,37,38,39,40,41,42]. In one study, the prevalence of *Pipt* in leaf-blast-resistant lines was reported [43] while significant association of *Pi56(1)* and *pi21* explained blast resistance in another study [44]. Genetic frequencies of 18 blast resistance genes *Pib*, *Piz-t*, *Pik*, *Pik-p*, *Pikm*, *Pik-h*, *Pita/Pita-2*, *Pi2*, *Pi9*, *Pi1*, *Pi5*, *Pi56(t)*, *Pia*, *Pi65(t)*, *Pi33*, *Pit*, *pi21* and *Pish* ranged from 6% to 27% in 288 landraces collected from northeastern India [45]. Another report stated the genetic frequencies of five major blast resistance genes to be in the range of 25 (*Pi1*) to 90.6% (*Pi2*) [46]. The majority of the previously developed blast resistant rice varieties have a single resistance gene, viz., *Pi2* [47] or *Pi9* [48] or *Pi54* [49].

### 4.2. Morphological Characterization

In the present investigation, when the 41 selected ILs were evaluated for key agro-morphological traits, many lines showed superiority over the original recurrent parent for various agro-morphological traits, and more particularly for plot yield. Owing to the complex nature of agronomic traits, understanding the genetic variability present in the genotypes under study, the factors contributing to genetic variability and the interactions among the component traits are of utmost importance. We employed various selection criteria in the identification of promising ILs possessing a desirable combination of component yield traits. ANOVA revealed the presence of significant variability for the yield traits in both seasons, and correlation analysis signified the positive association of grain yield with days to 50% flowering, tiller number and panicle number. Generally, highly significant negative correlation is observed between plant yield and plant height, which was also observed in the present study. Contrary to this, a significant positive correlation between yield and plant height was reported in other studies [50,51,52,53].

Principal component analysis (PCA), a multivariate technique, reduces data with a large number of correlated variables into a substantially smaller set of new variables through a linear combination of the variables that accounts for most of the variation present in the original variables. PCs explain the variability which could not be attributed to the other factors [50]. In the present investigation, the grouping of the ILs was mainly explained by the three major PCs, while a similar amount of cumulative variance with two to four major PCs was reported [50,54]. PCA biplots visualized the positive and negative associations among the component yield traits in the present study, suggesting the selection criteria for higher grain yield among the ILs. In the PCA plots, it is interesting to observe a clear-cut grouping of the ILs, although one with a few overlaps based on the recurrent parent’s background, both in WS and DS. PCA plots provided pictorial representations of phenotypic expression of the yield traits, which was promising in DS and poor in WS, in the pooled analysis.

Grain yield was poorly expressed in almost all the genotypes in WS, with an average yield of 439 g/m^2^. Though the season experienced very good rainfall during the crop growing period, the recorded low yields could be due to late sowings in the 2nd week of July and delayed planting of over-aged seedlings (>40 days old) resulting in poor tillering. Whereas in DS, a high average yield of 609 g/m^2^ was recorded. The differential yield levels in the present study may not truly reflect the seasonal effect, as timely sowings/plantings were not taken up during the wet season. In WS, a yield of 23 ILs was comparable to those of yield checks, and IL 19396 with blast resistance was the top ranked, with yield advantage of 9.57% over the best check. In DS, the yield of all the ILs was comparable to those of the checks, and 13 ILs recorded a higher yield, with yield advantage ranging from 3.1 to 33.1%. Similarly to the present study, seven ILs with high yield and blast resistance were identified in the background of Samba Mahsuri [55]. Multi-parent selective inter-crossing in the present study not only led to the development of high-yielding resistant ILs but also enhanced the recovery of the recurrent parent via selection for essential morphological features. Common genetic backgrounds from elite parents in each intercross, coupled with reliable phenotyping and screening at field level, enabled us to reject plant types differing from the RP, so only those ILs resembling the RP in maximum essential morphological traits were considered. This study is in agreement with previous studies that used agro-morphological based plant selection strategies in selecting breeding lines [13,14,15].

### 4.3. Background Selection

Generally, backcrossing of the target-trait positive plants to the recurrent parent is the norm in the development of near-isogenic lines. Although backcross breeding has been demonstrated to be an effective approach for improving two or three complex traits and the development of NILs in the elite cultivar background, the multiple trait improvement was mostly sequential, as has been reported earlier for drought and submergence in the literature by Swarna and Samba Mahsuri [56], bacterial blight resistance and low soil P tolerance in Samba Mahsuri [9], and bacterial blight resistance and salinity tolerance in [10]. In the present study, backcross breeding was not followed; instead, selective inter-crossing was practiced, involving multiple donors in order to accumulate the maximum number of genes in a common background, which also achieved enhanced recovery of the recurrent parent’s morphological features. To validate our findings based on essential morphological traits, we performed background selection in a set of ILs. Elite parent background recovery assessed by high throughput genotyping technique, such as genotyping by sequencing as used in the present study, confirmed the success of repeated cycles of selective intercrossing in the development of NILs, as the background recovery in the NILs varied from 90.2 to 93.76%. Earlier, we reported a 73.32 to 96.43% background recovery in the ILs of Krishna Hamsa using simple sequence repeat (SSR) markers [11]. Recent simulation studies have demonstrated that the inclusion of intercrossing in a trait introgression breeding program can substantially increase efficiency and the probability of success using fewer resources compared with backcross breeding strategies [17].

The novel feature of this study was a successful demonstration of multi-trait introgression in a single genetic background through a multi-parent selective inter-crossing program using phenotypic selection strategy validated by SNP-based background selection. The availability of these ILs and NILs possessing genes in elite backgrounds offers a valuable genetic resource in its combination of a higher number of genes with minimal linkage drag, which is a major limitation while using traditional donors in MAS. The high-yielding ILs and NILs with resistance to the most important rice diseases identified in the present study have been nominated for multi-location national trials across the country in the “All India Coordinated Research Project on Rice” (AICRPR). The promising lines with yield advantages over the checks will be released as varieties and will aid in ensuring the country’s food security.

## 5. Conclusions

The bacterial blight resistance in the ILs with missing targeted alleles was due to the presence of either *xa8* or *Xa38*, and blast resistance was due to *Pi1*, *Pi38*, *Pi40*, *Pi20*, *Pib* or *Pipt*, in various combinations. The present study also resulted in the identification of high-yielding disease resistant ILs and demonstrated the development of NILs through multi-parent selective intercrossing, which possessed more than 90% genetic similarity with the recurrent parent.

## Figures and Tables

**Figure 1 plants-12-03012-f001:**
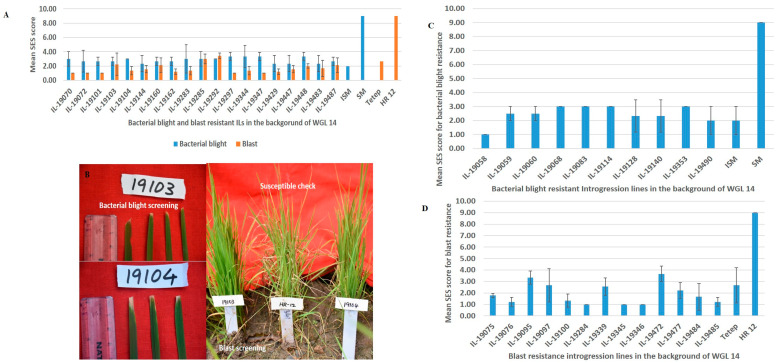
Bacterial blight and blast screening of introgression lines in the background of WGL 14 (RP2). Bacterial blight screening was performed in glass house in 2019 and 2020, and field screening in the wet season of 2020. Blast screening was done in universal blast nursery (UBN) during the wet seasons of 2018, 2019 and 2020. A score of ≤3 on the standard evaluation system (SES) scale is considered to indicate resistance to both bacterial blight and blast. ISM Improved Samba Mahsuri as resistant check, recorded an SES score of 2, and SM-Samba Mahsuri as susceptible check, recorded a score of 9 for bacterial blight; Tetep as resistant check, recorded a score of 2.67, and HR-12 as susceptible check, recorded an SES score of 9 for blast. (**A**) RP-2 ILs with resistance to both bacterial blight and blast. (**B**) Representative photographs of RP2-ILs 19103 and 19104 with bacterial blight resistance (on left side) and blast resistance (on right side). HR12 is the susceptible check used in UBN. (**C**) RP-2 ILs with resistance to bacterial blight. (**D**) RP-2 ILs with resistance to blast.

**Figure 2 plants-12-03012-f002:**
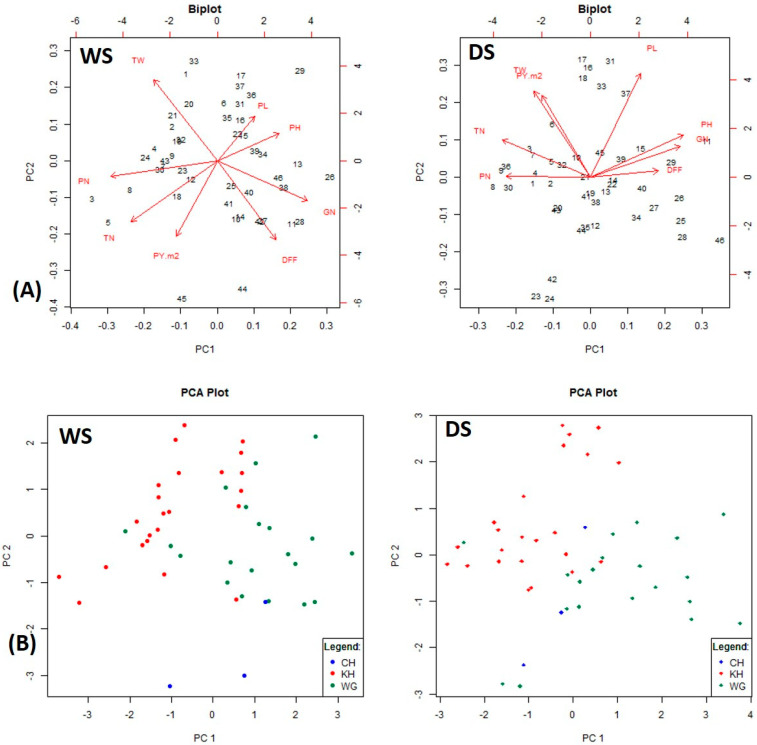
Principal component analysis (PCA) for eight yield traits among 46 rice genotypes. (**A**) PCA biplot depicting contribution of yield traits and genotypes. (**B**) PCA plot depicting distribution of the 46 genotypes. Blue dots represent checks, red dots represent ILs in the background of ‘Krishna Hamsa’ and green dots represent ILs in the background of WGL 14.

**Figure 3 plants-12-03012-f003:**
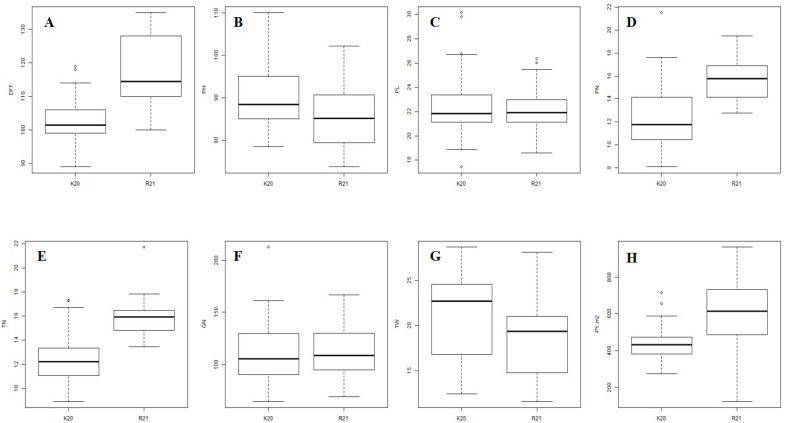
Box plots showing variation among the two seasons for yield traits. K20 represents the wet season (kharif) of 2020, and R21 represents the dry season (*rabi*) of 2021. The upper, median and lower quartiles represent the 75th, 50th and 25th percentiles of the introgression lines, respectively. The vertical lines represent the variation in the population. Dots represent the outliers. (**A**) Box plots for days to 50% flowering (DFF). (**B**) Box plots for plant height (PH) in cm. (**C**) Box plots for panicle length (PL) in cm. (**D**) Box plots for tiller number (TN). (**E**) Box plots for panicle number. (**F**) Box plots for grain number (GN). (**G**) Box plots for thousand grain weight (TW) in g. (**H**) Box plots for plot yield (PY) in g/m^2^.

**Table 1 plants-12-03012-t001:** Marker positivity for the bacterial blight and blast resistance genes in the introgression lines with resistance to both BB and BL.

S. No	IL	Gene Combination	S. No	IL	Gene Combination
1	19007	*xa5 + Pi1 + Pi20* + *Pi40* + *Pipt*	15	19144	*xa5 + Xa21 + xa8 + Pi54 + Pi38*
2	19019	*xa5* + *Xa21* + *Pi1 + Pi20* + *Pipt*	16	19160	*xa8 + Pi54* + *Pi20 + Pipt*
3	19020	*xa5* + *Pi1 + Pi20* + *Pi38*	17	19162	*xa5 + Xa38 + Pi54* + *Pi1 + Pi38* + *Pipt*
4	19025	*xa5 + xa8* + *Pi20* + *Pi38 + Pipt*	18	19283	*xa5 + Xa21 + xa8 + Pi9 + Pi1* + *Pib*
5	**19030**	***xa5* + *Xa21* + *xa8* + *Pi54* + *Pi1* + *Pi20* + *Pi38* + *Pipt* + *Pib***	19	19285	*xa5* + *xa8* + *Pi9* + *Pi1* + *Pi20* + *Pipt*
6	19246	*Xa38* + *Pi1 + Pipt*	20	19292	*xa5 + xa8 + Pi9* + *Pi20 + Pipt*
7	19378	*xa5 + xa13* + *Xa21 + xa8* + *Pi1* + *Pi20 + Pipt*	21	19297	*xa5* + *Xa21 + xa8 + Pi9 + Pi38* + *Pib*
8	19379	*xa5 + Pi54 + Pi1*	22	**19344**	** *xa5 + xa13 + Xa21 + xa8 + Pi9 + Pi1 + Pi20 + Pipt* **
9	19031	*xa5* + *Xa21 + xa8* + *Pi1 + Pi38 + Pib*	23	**19347**	** *xa5 + xa13 + Xa21 + xa8 + Pi9 + Pi1 + Pi20 + Pipt* **
10	19070	*xa5 + Xa21 + xa8* + *Pi38 + Pipt*	24	19429	*xa5* + *Xa21 + xa8 + Pi54* + *Pi1 + Pi20*
11	19072	*xa5* + *Xa21 + xa8* + *Pi20 + Pipt*	25	19447	*xa5 + Xa21 + xa8 + Pi9* + *Pi20 + Pipt*
12	19101	*Xa21 + xa8* + *Pi1* + *Pi20 + Pi38* + *Pib*	26	19448	*xa5 + xa8 + Pi9* + *Pi1 + Pi20* + *Pi38* + *Pipt*
13	19103	*xa8 + Pi9* + *Pi1* + *Pi20 + Pi38*	27	19483	*xa5* + *Xa21 + xa8* + *Pi20* + *Pi1 + Pipt* + *Pib*
14	19104	*xa5 + Xa21 + xa8* + *Pi20 + Pi38*	28	19487	*xa5* + *Xa21 + xa8 + Pi54*

ILs numbered from S. No 1 to 9 are in the background of Krishna Hamsa, and ILs numbered from S. No 10 to 28 are in the background of WGL 14. ILs with an underline are without targeted gene introgressions. ILs in bold possess maximum resistance gene introgressions.

**Table 2 plants-12-03012-t002:** Marker positivity for the bacterial blight genes in the introgression lines with resistance to bacterial blight.

S. No	IL	Gene Combination	S. No	IL	Gene Combination
1	19039	*xa5 + xa8*	15	19076	*xa5 + Xa21 + xa8*
2	19046	*xa5 + xa8 + Xa38*	16	19095	*xa5 + xa8*
3	19232	*xa5 + xa8 + Xa38*	17	19097	*xa5 + xa8*
4	19233	*xa8 + Xa38*	18	19100	*Xa21 + xa5 + xa8*
5	19238	*xa5 + Xa38 + xa8*	19	19284	*Xa21 + xa8*
6	19239	*xa5 + Xa38 + xa8*	20	19339	*xa5 + xa8*
7	19240	*xa5 + xa8 + Xa38*	21	19345	*xa5 + xa13 + xa8*
8	19244	*xa5 + xa8*	22	**19346**	** *Xa21 + xa13 + xa5 + xa8* **
9	19245	*xa5 + Xa38*	23	19472	*xa5 + xa8*
10	19247	*xa8*	24	19477	*xa5 + Xa21 + xa8*
11	19248	*xa5 + Xa38*	25	19484	*Xa21 + xa5 + xa8*
12	19406	*xa5 + xa8*	26	19485	*Xa21 + xa5 + xa8*
13	19460	*xa5 + Xa21 + xa8*	27	**19075**	** *Xa21 + xa5 + xa8 + Xa38* **
14	**19379**	** *Xa21 + xa5 + xa13 + xa8* **			

ILs numbered from S. No 1 to 13 are in the background of Krishna Hamsa, and ILs numbered from S. No 14 to 27 are in the background of WGL 14. ILs with an underline are without targeted gene introgressions. ILs in bold possess the highest number of resistance gene introgressions.

**Table 3 plants-12-03012-t003:** Marker positivity for the blast resistance genes in the introgression lines with resistance to blast disease.

S. No	IL	Gene Combination	S. No	IL	Gene Combination
1	**19013**	** *Pi9 + Pi54 + Pi1 + Pi20 + Pi40* **	19	19214	*Pi9 + Pi1 + Pi38 + Pipt*
2	19015	*Pi54 + Pi1 + Pi20*	20	19396	*Pi1 + Pi20*
3	19016	*Pi9 + Pi1 + Pi20 + Pi38*	21	19408	*Pi9 + Pi1 + Pi20*
4	19018	*Pi1 + Pi20 + Pi38*	22	19411	*Pi20 + Pib*
5	19021	*Pi1 + Pi38*	23	19420	*Pi9 + Pi20 + Pib*
6	19022	*Pi9 + Pi1 + Pi20*	24	19421	*Pi9 + Pi1 + Pi20*
7	19023	*Pi1 + Pi20*	25	19461	*Pi9 + Pi1 + Pi20*
8	19024	*Pi1 + Pi20*	26	19471	*Pi9 + Pi1 + Pi20*
9	19026	*Pi1 + Pi20 + Pi38 + Pipt*	27	19058	*Pi1 + Pi20*
10	19033	*Pi54 + Pi1 + Pi38 + Pipt*	28	19059	*Pi1 + Pi20*
11	19042	*Pi54 + Pi1 + Pi20 + Pi38*	29	19060	*Pipt + Pi1 + Pi20*
12	19180	*Pi1 + Pi20*	30	19068	*Pi38 + Pib*
13	19181	*Pi1 + Pi20 + Pi38*	31	19083	*Pi54*
14	19182	*Pi9 + Pi1 + Pi20 + Pi38*	32	19114	*Pi54 + Pipt + Pib*
15	**19185**	** *Pi9 + Pi1 + Pi20 + Pi40 + Pipt* **	33	**19128**	** *Pi54 + Pipt + Pib + Pi1 + Pi20 + Pi38* **
16	19206	*Pi9 + Pi1 + Pipt*	34	19140	*Pi54 + Pi38 + Pipt*
17	19207	*Pi9 + Pi1 + Pipt*	35	*19353*	*Pi9 + Pi1 + Pi20 + Pi38 + Pipt*
18	19211	*Pi1 + Pi38 + Pipt*	36	19490	*Pi20*

ILs numbered from S. No 1 to 26 are in the background of Krishna Hamsa and ILs numbered from S. No. 27 to 36 are in the background of WGL 14. ILs with an underline are without targeted gene introgressions. ILs in bold possess the greatest number of resistance gene introgressions.

**Table 4 plants-12-03012-t004:** Environment-wise and pooled analysis of variance (ANOVA) of yield-contributing traits in the introgression lines.

Source	df	MSS (WS 2020)	MSS (DS 2021)	df	MSS (Pooled)
	Days to 50% flowering (DFF)				
Replication	1	0	0	1	0
Genotypes	45	98	205	45	214 ***
Error	45	0	0	137	95
Total	91			183	
Plant height (PH) in cm					
Replication	1	42	1	1	29
Genotypes	45	127 ***	130 ***	45	219 ***
Error	45	25	5	137	30
Total	91			183	
Panicle length (PL) in cm					
Replication	1	7	0	1	5
Genotypes	45	20 ***	8 ***	45	20 ***
Error	45	4	1	137	4
Total	91			183	
Tiller number (TN)					
Replication	1	0	1	1	1
Genotypes	45	24 ***	12 **	45	18 **
Error	45	8	8	137	15
Total	91			183	
Panicle number (PN)					
Replication	1	1	1	1	2
Genotypes	45	35 ***	12 ***	45	29 ***
Error	45	11	5	137	15
Total	91			183	
Grain number (GN)					
Replication	1	157	116	1	2
Genotypes	45	2766 ***	1714 ***	45	2684 ***
Error	45	783	132	137	896
Total	91			183	
Thousand grain weight (TW) in g					
Replication	1	7	0	1	2
Genotypes	45	48 ***	35 ***	45	70 ***
Error	45	3	1	137	9
Total	91			183	
Grain yield/Plot yield (PY) in g/m^2^					
Replication	1	611	810	1	7
Genotypes	45	19234 ***	78207	45	57117 ***
Error	45	6016	7077	137	27300

df—Degrees of freedom, MSS—Mean Sum of Squares, ** indicates *p* value at 0.01, and *** at the 0.001 level of significance.

**Table 5 plants-12-03012-t005:** Correlation of yield traits in 46 rice genotypes.

	DFF	PH	PL	TN	PN	GN	TW	PY (g/m^2^)
Wet season 2020								
DFF	1	0.38 **	0.86 ***	0.74 ***	0.16	0.55 ***	0.05	−0.13
PH		1	0.17	−0.23	−0.23	0.31 *	−0.18	0.03
PL			1	−0.16	−0.14	0.23	0.11	−0.15
TN				1	0.66 ***	−0.23	0.16	0.30 *
PN					1	−0.48 ***	0.27	0.29 *
GN						1	−0.31 *	−0.10
TW							1	−0.05
Dry season 2021								
DFF	1	1.00 ***	1.00 ***	1.00 ***	1.00 ***	0.30 *	0.75 ***	−0.03
PH		1	0.34 *	−0.18	−0.27	0.47 ***	−0.27	−0.10
PL			1	−0.06	−0.22	0.44 **	0.33 *	0.17
TN				1	0.45 ***	−0.44 **	0.32 *	0.12
PN					1	−0.03	−0.05	0.39 **
GN						1	−0.20	0.08
TW							1	0.31 *
py								1
Pooled Correlations								
DFF	1	−0.03	−0.01	0.40 ***	0.24 *	0.23 *	−0.41 ***	0.32 **
PH		1	0.30 **	−0.36 ***	−0.40 ***	0.32 ***	−0.03	−0.23 *
PL			1	−0.14	−0.18	0.26 *	0.23 **	−0.02
TN				1	0.77 ***	−0.17	−0.10	0.52 ***
PN					1	−0.25 *	−0.06	0.52 ***
GN						1	−0.34 ***	0.02
TW							1	−0.04

DFF—Days to 50% flowering, PH—Plant height, PL—Panicle length, TN—Tiller number, PN—Panicle number, GN—Grain number, TW—Thousand grain weight, PY—Plot yield. * indicates *p* value at 0.05, ** at 0.01, and *** at the 0.001 level of significance.

**Table 6 plants-12-03012-t006:** Environment and pooled PCA analysis of the yield traits in 46 rice genotypes.

Variables	PC1	PC2	PC3	PC4	PC5	PC6	PC7	PC8
Wet season 2020								
DFF	28.65	−48.13	10.11	−46	23.91	−27.9	−52.63	22.9
PH	30.59	16.71	39.22	63.8	50.4	−12.03	−20.85	−7.49
PL	18.35	27.19	70.76	−33.19	−31.02	−32.36	27.04	−8.6
TN	−42.44	−37.3	32.92	−15.27	29.74	23.88	3.95	−63.41
PN	−52.52	−9.43	26.87	3.24	30.68	−5.64	28.85	67.93
GN	44.43	−24.5	24.99	−0.53	−8.48	76.48	17.79	23.72
TW	−31.13	49.66	20.64	−15.16	−10.77	37.68	−65.19	11.07
PY	−20.1	−46.02	22.26	47.31	−62.54	−12.57	−25.49	5.03
Standard deviation	1.59	1.28	1.04	0.92	0.83	0.72	0.69	0.48
Proportion of variance	31.53	20.33	13.57	10.66	8.6	6.49	6	2.82
Cumulative proportion	31.53	51.86	65.43	76.09	84.69	91.18	97.18	1
Eigen values	2.52	1.63	1.09	0.85	0.69	0.52	0.48	0.23
Dry season 2021								
DFF	32.16	3.7	14.41	−71.34	47.57	−35.77	2.15	10.4
PH	44.12	24.93	7.33	−35.64	−41.59	47.59	−17.91	−42.33
PL	23.7	60.72	−14.4	18.72	−16.57	−36.16	59.64	−7.4
TN	−42.16	22.1	−3.49	−42.48	−56.57	−14.01	−11.4	48.89
PN	−40.05	0.73	64.05	−3.44	−10.83	−33.59	1.5	−55.08
GN	42.54	18.33	42.7	38.2	−10.34	−26.09	−51.84	33.11
TW	−26.93	50.59	−42.39	2.62	30.01	−13	−54.96	−28.67
PY	−23.27	47.89	42.39	2.6	37.51	54.53	16.68	26.5
Standard deviation	1.60	1.29	1.09	0.96	0.88	0.68	0.53	0.38
Proportion of variance	31.91	20.91	14.72	11.61	9.67	5.83	3.51	1.83
Cumulative proportion	31.91	52.82	67.54	79.15	88.83	94.66	98.17	100
Eigen values	2.55	1.67	1.18	0.93	0.77	0.47	0.28	0.15
Pooled PCA								
DFF	28.21	−49.73	−2.37	27.44	45.98	−56.83	−13.5	21.13
PH	−34.91	−30.77	18.9	73.58	−44.46	8.96	−2.55	0
PL	−20.02	−18.08	72.36	−10.5	44.5	24.71	36.53	−1.12
TN	52.81	−3.72	15.69	20.52	8.79	26.7	−28.98	−70
PN	52.15	7	13.15	13.55	−7.73	46.93	−10.37	67.12
GN	−15.31	−59.33	7.91	−50.76	−21.11	16.75	−53.49	4.37
TW	−11.33	49.97	52.95	2.08	−4.01	−37.03	−55.55	9.64
PY	41.38	−13.14	33.41	−23.19	−57.71	−38.9	39.78	−5.94
Standard deviation	1.66	1.32	1.10	0.79	0.76	0.71	0.62	0.46
Proportion of variance	34.33	21.88	14.99	7.78	7.3	6.23	4.86	2.63
Cumulative proportion	34.33	56.21	71.2	78.98	86.28	92.51	97.37	100
Eigen values	2.75	1.75	1.20	0.62	0.58	0.50	0.39	0.21

PC—Principal components, DFF—Days to fifty-percent flowering, PH—Plant height (cm), PL—Panicle Length (cm), TN—Tiller Number, PN—Panicle Number, GN—Grain Number, TW—Thousand grain weight (g), PY—Plot yield (g/m^2^).

**Table 7 plants-12-03012-t007:** Identification of near-isogenic lines with resistance to bacterial blight and/or blast in the background of two elite rice cultivars, ‘Krishna Hamsa’ and ‘WGL14’.

IL	Gene Combination	Resistant to	DFF			PH (cm)			Plot Yield (g/m^2^)			RPG %
			WS 2020	DS 2020	Pooled	WS 2020	DS 2020	Pooled	WS 2020	DS 2020	Pooled	
19020	*xa5* + *Pi20 + Pi38 + Pi1*	BB & BL	102	116	109	89	79	84	476	780	628	91.90
19185	*xa5* + *Pi9 + Pi20 + Pi40 + Pi1 + Pipt*		97	113	105	87	80	83	515	501	508	92.50
19206	*xa5 + Pi9* + *xa8 + Pipt + Pi1*		98	108	103	92	76	84	399	417	408	92.51
19026	*xa5* + *xa8 + Pi20 + Pi38 + Pipt* + *Pi1*	BL	101	101	101	81	74	78	463	671	567	90.70
19211	*Pi38 + Pipt + Pi1*		94	108	101	84	79	81	422	509	466	92.19
19396	*xa5* + *Pi20 + Pi1*		97	112	105	84	79	81	608	687	647	91.58
**Krishna Hamsa**		**--**	**98**	**105**	**102**	**84**	**77**	**80**	**473**	**474**	**474**	
19345	*xa5 + xa13* + *Pi9* + *xa8 + Pi1 + Pipt + Pi20*	BB & BL	119	128	124	88	90	89	308	471	390	93.26
19484	*xa5* + *Xa21 + Pi54* + *xa8* + *Pi1 + Pib + Pipt*		101	128	115	98	88	93	426	905	666	91.81
**WGL14**		**--**	**111**	**129**	**120**	**95**	**92**	**94**	**427**	**490**	**455**	

IL—Introgression line; DFF—Days to 50% flowering; PH—Plant height in cm; PY—Plot yield in g/m2; RPG %—Recurrent parent genome recovery; WS—Wet season; DS—Dry season; BB—Bacterial blight; BL—Blast.

## Data Availability

Not applicable.

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
