# Peer review of "Molecular and Morphological Characterization of Introgression Lines with Resistance to Bacterial Leaf Blight and Blast in Rice"

_plants, 2023, doi:10.3390/plants12163012_

Round 1
Reviewer 1 Report (New Reviewer)
The present study focuses on the marker-assisted forward breeding derived introgression lines with resistance to bacterial blight and/ or rice blast disease. The data showed that multi-parent intercrossing could develop high-yielding disease-resistant lines.
The paper is missing some key information. As for Bacterial blight or rice blast screening, which pathogen strains were used in this paper respectively? Please list their names and related information in this paper. And the inoculation methods used for both pathogens should also be added.
Author Response
The present study focuses on the marker-assisted forward breeding derived introgression lines with resistance to bacterial blight and/ or rice blast disease. The data showed that multi-parent intercrossing could develop high-yielding disease-resistant lines.
The paper is missing some key information. As for Bacterial blight or rice blast screening, which pathogen strains were used in this paper respectively? Please list their names and related information in this paper. And the inoculation methods used for both pathogens should also be added.
Response: Thank you for the suggestion. The details of the disease screening, method of inoculation and pathogen strains are included under material and methods section in the revised manuscript.
Reviewer 2 Report (New Reviewer)
Tables and figures
· For all the figures please use insert caption to separate text from the figures and tables. Align the figures and tables and the text nicely.
· In Table 1, Table 2, Table 3: what does S. No denote?
· Table 1, Table 2, Table 3: “ILs with an underline are without targeted gene introgressions.” Cannot tell in the current state of the manuscript since everything is underlined
· Table 4 and Table 5: what do *, **, *** denote?
· Figure 2 is too small! – not clear. Please enlarge.
Materials and methods
· The phenotyping was not illustrated and information of whether screening was done in the glasshouses was not provided in the material and methods section, it only appears at Fig 1.
· Materials and methods, results and discussion to be clarified and to be aligned with the aim of the study
· The different subheadings to have a logical flow on how the experiment started, which material was used, how were they developed, where and how were they grown, which molecular markers were used…properly categorize the materials with the methods and results, and discussions.
· Allocate subheadings for discussions and discuss data in comparison to previous findings.
Conclusion
· Clarify the conclusion to answer the aim of the study in line with the findings…shorten the conclusion and consider moving some of the statements in the discussion session. The conclusion should clearly provide an indication which markers are associated with resistance in the MAFB-ILs…
References
· Please check reference no: 46
P1L6: remove and before Raman
P1L18: replace focuses on with evaluates
P1L20: rephrase introgression of none/few, the message is not delivered clearly. Do the R genes mean resistance genes? It is not explained anywhere in the document
P1L23: is it 20 R genes or markers? How does genes reveal the presence of other genes! Provide consistency on writing xa8 and Xa38 and other genes throughout the manuscript
P1L26: replace was with ‘were’ observed
P1L27-28: thousand grain weight is also significantly and positively correlated with grain yield. On the contrary, grain yield showed a significantly negative association with plant height.
P1L28: Add fifty percent ‘(50%)’ flowering or try an easy to follow and consistent abbreviation
P1L30: 50% flowering
P1L37-39: Too many keywords. Avoid using keywords same as the title
P2L47-60: You talk about resistance and mechanisms of control, then you introduce the BB and BL, then you are back at the modes of control. The ideas are all over the place. Rephrase like this:
Line 57-58 should be moved to line 47. Then line 53-57 should follow after that. Still on that, extend your synthesis in line 53-54 by giving examples of BB effect at for example, seedling, tillering, flowering, heading and stage or on agro-morphological traits.
It would have been nice to give a little background of what causes Bacterial Blight like that of the blast disease
The aim of the study is not clear or must be rephrased to make it understandable to a reader with no background of molecular studies
P2L48: The statement about host resistance is not true
P2L49-50: rewrite the sentence. Is it the breakdown of varieties or the breakdown of resistance?
P2L55: BB-Resistance genes? (BB-R)
P2L57: Write the fungus name in full by adding Magnaporthe ‘oryzae’.
P2L58-59: Repetition of line 53-55
P1L47-52: Should be moved to line 60 just before the sentence ‘to date, more than 100’……
P2L58: space between grey/ and white
P2L62: Add ‘(MAS)’ after Marker-assisted breeding
P2L62: substitute ‘had’ with ‘has’. Delete in the recent past both
P2L66: substitute ‘in’ with across different
P2L71: delete ‘s’ in observations
P2L74: L74: Rephrase the aim of the study to clarify…you may consider mentioning the aim of the study is to determine genes associated with resistance to BB or BL…as the aim does not specify what it seeks to do
P2L78-79: Add: This requires ‘multi-season’ field evaluations
P2L80: insert hyphen (-) for ‘sub-set’
P2L76-90: Consider moving to material and methods and brief discussion of results and fit it in the relevant sections or can be integrated to flow nicely in the introduction supported by references
P2L91: Results to be moved after material and methods section and results section should have a logical flow with various subheadings associated with the methods deployed. An example would be sections for molecular characterization in the same sequence it follows in material and methods section…thus, if it is the first bullet, then it would be ideal if it is the first bullet in the results and discussion section.
P2L92-94: Consider rephrasing like this ‘Of the 91 ILs evaluated in the present investigation…196 MAFB-ILs derived in the background of…and 43 ILs derived in the background of…’
P2L97: Add ’(BB)’ after bacterial blight. Also, maintain the consistency of writing the two diseases in abbreviation or in full throughout the whole manuscript and also writing numbers in full or as numbers full words for number e.g ten or 10
P3L110: Add ‘(BB) after bacterial blight and (BL)’ after blast since the abbreviations are used randomly in line 117
P4L125, 135 and 147: Why not bold the lines with maximum R gene introgressions to increase your emphasis
P4L137-138: There were ten RP1-ILs and five RP2-ILs
P5L151: …and dry season 2021
P5L151: Abbreviation ‘(WS)’ for wet season should be added as in line 156.
P5L151: Abbreviation ‘(DS)’ for dry season should be added as in line 157
P5L154: please correct the typo ‘per cent’ to ‘percent’ and if fifty percent to flowering is abbreviated as DFF why not use the abbreviation in the abstract as well.
P5L154: …except for days to fifty percent flowering…
P5L154-162: Please check the usage of these abbreviations in the abstract and the rest of the document
P5L162: Noteworthy, DFF showed significantly positive correlations with all traits, except for plot yield in the dry season 2021. The significantly positive correlations were also observed with most traits, except panicle number, thousand grain weight and plot yield in the wet season 2021. The same trend was also observed in the pooled analysis, except that DFF showed no significant associations with plant height and panicle length, while a significantly negative association was noted between DFF and thousand grain weight.
P6L164-168: Please indicate the correlation at what level ‘0.01, 0.0001’ by which asterisk
P7L171: rephrase this sentence
P7L171: The PCA extracted principal components (PCs) in both the seasons as well as in the pooled analysis.?
P7L175-177: The positive effect of the variables DFF, PH, PL and GN and the negative effect of the variables TN, PN, TW and PY for PC1 were commonly observed…(Table 6).
P7L181: Add ‘the’ rest of the genotypes
P7L182: Remove ‘of’ after alongside
P8L189: Fig. 2 is blurry. Try to replot it with visible legends and axis
P8L198: …was observed in the DS (Supplementary Table S2).
P9L209-210: Six were taller (which are? List them). Add “the” before the remaining
P9L217: See the previous comment
P9L219-220: The remaining lines were they similar in their values or their values were grouped in the same cluster (longer/shorter PL) as RP?
P9L227-228: The remaining lines were they similar in their values or their values were grouped in the same cluster (more/lesser TN) as RP?
P9L229: wide range of??
P9L237: …with outliers both in the wet and dry seasons (Fig 3F).
P9L244-245: The remaining lines were they similar in their values or their values were grouped in the same cluster (more/lesser GN) as WGL14?
P9L246: Wide range of what? Consider writing it this way ‘A wide range in TW was observed (Fig 3G), which varied from 11.9-29.45 g with a mean of 21±4.59 g in the wet season (Supplementary Table S1) and from 11.51-29.05 g with a mean of 18±4.01 g in the dry season (Supplementary Table S2).’
P9L249: 11.79 g…27.14 g… 20±4.54 g…
P9L250-252: Similar in values or in category (Higher/lower TW)?
P10L269: since background recovery is being reported on, it would be interesting to show at what stage the plants were in and if it was achieved much quicker with the use of molecular technology
P11L280-281: the statement must be elaborated so it doesn’t cause confusion
P11L282: substitute the word devoid
P11L285: in another study,…led not lead… gene/QTL
P11L288-290: Rephrase the sentence
P11L296: substitute and the increase with an increase
P11L298: “As many as 27 ILs were detected with missing allele of the targeted BL-R genes in the present study.”??? Was not 15 out of 36 ILs? (Table 3)?
P12L349: has this method been followed anywhere else? Please give references if so.
P12L358: Material and methods
P12L359: This sentence comes across as if though the material used was already resistant. Why was the resistant material used?
P12L360: insert “were” after blast
P12L361: insert space after 14’
P12L361-362: it is good to refer the reader to the previous work regarding the development ‘Krishna Hamsa’ ILs, however, the authors should give a brief description or a list of the donors that were involved in the development of ‘Krishna Hamsa’ ILs.
P12L365-369: Tetep with Pi9 and Pi54; RP Patho-1 with Pi2 for BL; Raathu Heenathi with Bph3 and Bph17 for brown plant hopper; RP 5924-366 24 with Gm4; RP 5924-25 with Gm8 for gallmidge resistance; IR 96321-1447-561-B-367 1 with qDTY 1.1 and qDTY 3.1; IR 81896-96-B-B-195 with qDTY 2.1; and…
P13L370: briefly describe the breeding strategy. See the previous comment
P13L383: delete “(check 1), (check 2), (check 3)” and write “…..Improved Samba Mahsuri (ISM), Samba Mahsuri, and Swarna were used as yield checks 1, 2 and 3, respectively or Additionally, three elite cultivars including Improved Samba Mahsuri (ISM) (check 1), Samba Mahsuri (check 2), and Swarna (check 3) were used as yield checks.
P13L385 & 386: delete “taken up” replace them with “conducted”.
P13L387: 8th February 2021… Experiments were…
P13L393: insert “was measured” after (PH)
P13L400 & 402: “in g” or “weighed in grams”, you can only mention it once in a sentence. after drying; …
P13L407: rephrase the sentence. For example…. Correlation and principal analyses were conducted on yield data. Or delete “other statistical parameters viz.,”
P13L420: Some genes not appearing in the material and methods are mentioned here. Were these genes evaluated as well? Maybe clearly indicate in the materials and methods section as to which targeted resistance gene were tested.
P14L427: …targeted Pi2, Pi9 or Pi54…
P14L434: virulent Xoo isolates?
P14L436: what about xa38, will xa8 be used alone or with the presence of other genes?
Author Response
Authors profusely thank the reviewer for the exhaustive review, critical and valuable suggestions. We attended to all the comments in the revised manuscript and response to the reviewers comments are attached in the file

Reviewer 3 Report (New Reviewer)
Line 2, Actually, the major descriptioninResults was Morphological characterization of 41 ILs, and it should be reflected in the title of the manuscript.
Line 108, How to know the R gene information in Table 1, and it should be described briefly.
Line 128,‘, however,’
Line 149, The description of ‘with desirable agronomic traits and phenotypic acceptability based on visual characteristics’ is indistinct, has the author considered the information of R-gene combination in Table 1 to Table 3?
Line 151, Is it ‘and dry season 2021’ ?
Line 190, The resolution of Fig.2 is low, please improve it.
Line 269,Actually,it seems to be independent of each other among the three parts in the manuscript: ‘2.1. Molecular characterization’, ‘2.2. Morphological characterization’, and ‘2.2.4. Background recovery’. The author should provide more relevant information to clarify their internal logical relationships.
Line 275, How to calculate the value of RPG%, what is the relationship between RPG and the three traits in Table 7: DFE, PH, and Plot yield?
Moderate editing of English language
Author Response
Authors greatly thank the reviewer for the constructive comments. All the comments have been attended in the revised manuscript and response to the comments is attached in the file

Reviewer 4 Report (New Reviewer)
In my observation minor correction is required where highlighted with yellow color.

Author Response
In my observation minor correction is required where highlighted with yellow color
Response: Thank you. Minor corrections have been made that were highlighted with yellow color
Reviewer 5 Report (New Reviewer)
Although the manuscript was well-revised, there are still some concerns need to be addressed.
1. In ‘2. Material and Methods’, a subtitle may be required for the materials. Besides, please make sure if the ‘check’ means ‘control’, such as in the ‘resistant check’ and ‘susceptible check’.
2. Should the ‘don’ be ‘do’ at line 20?
3. Ref 13 is missing. Ref is required for the sentence at lines 565-566.
4. Please specify ‘BPH’ at line 146, RPG at line 492.
5. For the blast fungus, both fungus ‘Magnaporthe oryzae (line 57)’ and ‘Pyricularia oryzae (line 173)’ were used.
6. The pictures and figure legend in Figure 1 and Figure 2 should be reorganized. Please specify ‘SES’ in Figure 1.
7. A subtitle is required for the first part of the results. Alternatively, this part could be described in ‘Material and Methods’.
8. It seemed like the molecular characterization and the morphological characterization were performed separately. Could the authors addressed more on the relationship between them?
Minor revision is required to avoid ambiguity.
Author Response
Although the manuscript was well-revised, there are still some concerns need to be addressed.
Response: Authors thank the reviewer for the suggestions. Authors would like to bring to the notice of the reviewer #5 that most of the suggestions made were addressed in the earlier version, however, it appears that the first revised version was sent to the reviewer #5, instead of the latest revised version submitted by the authors. Hence, authors considered the last revised version with no track changes and additional suggestions made by the reviewer #5 were addressed and highlighted in yellow color/red font.
- In ‘2. Material and Methods’, a subtitle may be required for the materials. Besides, please make sure if the ‘check’ means ‘control’, such as in the ‘resistant check’ and ‘susceptible check’.
Response: Subtitle is added now. Now the “control” term is added along with checks when it is used in M& M for the first instance.
- Should the ‘don’ be ‘do’ at line 20?
Response: This was corrected in the earlier version, now highlighted in yellow
- Ref 13 is missing. Ref is required for the sentence at lines 565-566.
Response: Reference 13 is highlighted in the text at line nos. 75, 82 and 95
- Please specify ‘BPH’ at line 146, RPG at line 492.
Response: Specified
- For the blast fungus, both fungus ‘Magnaporthe oryzae (line 57)’ and ‘Pyricularia oryzae (line 173)’ were used.
Response: Magnoporthe oryzae and Pyricularia oryzae are invariably used for referring blast fungus. Now in line 173 it is changed to Magnaporthe oryzae and made uniform throughout the manuscript
- The pictures and figure legend in Figure 1 and Figure 2 should be reorganized. Please specify ‘SES’ in Figure 1.
Response: Figure 1 and Figure 2 are reorganized
- A subtitle is required for the first part of the results. Alternatively, this part could be described in ‘Material and Methods’.
Response: Subtitle is included and section numbers under M& M changed accordingly
- It seemed like the molecular characterization and the morphological characterization were performed separately. Could the authors addressed more on the relationship between them?
Response: Molecular and morphological characterization are carried out separately. The relationship between them is shown in second last para of introduction, data is showed in Tables, 1, 2, 3 & 7 and also explained thoroughly under results section 3.1.1, 3.1.2 and 3.1.3 and under discussion
Round 2
Reviewer 2 Report (New Reviewer)
This version is better than the previous version.
OKAY
Author Response
Authors thank the reviewer for the valuable suggestions which greatly improved the quality of the manuscript
Reviewer 3 Report (New Reviewer)

Moderate editing of English language required
Author Response
Response to Reviewer’s comments
Authors thank the reviewer for the suggestions. All the comments made by the reviewer have been addressed and response to each comment is given below.
Line 53-57, There is no need to describe the symptoms of rice caused by Xanthomonas oryzae pv. Oryzae.
Response: The symptoms were described in the earlier revised version in response to the comments made by Reviewer 2.
Line 122, What does BB and BL stand for?
Response: The abbreviated forms ‘BB’ and ‘BL’ are removed and their full terms are included now in the revised manuscript.
Line 221, The resolution of Fig.1 A, C and D is low, please improve it. I can’t distinguish numbers of the ruler in D.
Response: Thank you for the suggestion. The figures are now formatted with better resolution and the original JPEG images are submitted as separate files now
Line 310, There is only 1 replication, why?
Response: Since the number of replications is 2, the degrees of freedom is 1.
Line 341, The resolution of Fig.2 A is low
Response: The original JPEG images are submitted as separate files now
This manuscript is a resubmission of an earlier submission. The following is a list of the peer review reports and author responses from that submission.
Round 1
Reviewer 1 Report
This manuscript described how 91 ILs were developed and gained more resistance to BB, BL, and Bph, and superior agronomic traits. However, the authors have failed in multiple aspects of materials and methods and results.
1, Materials and Methods: No details about how genotyping and marker-assisted selection was reported. If there was no MAS, the authors must explain why.
2. Results: The authors tried not to explain the results. Instead, DNA extraction protocol was repeated in Lines 501 - 545. This must be removed to materials and methods.
3. Results: The authors again explained how experiments were conducted, between Line 548 - 562. This should be removed to M&M.
4. Results: The authors again described how their results were analysed. Lines 564 - 581 must be transferred to materials and methods.
5. Results: The authors must describe the results and discuss them completely.
Reviewer 2 Report
The current manuscript provides the results of intensive and years-long work of pyramiding resistance genes and QTLs. The work provides important breeding materials, which is an important achievement. The main issues are the quality of writing, figures, and tables. For examples, the abstract and the main body are so tedious, needs to be concise and to the point; not clearly representing the data, the legends are not clearly describing all the terms; The information of the markers are missing.
Minor issues
Line 21 Marker-assisted multiparent-derived introgression lines (ILs)
Line 32 in targeted should be the targeted
Line 66 check this sentence
Line93-96 rewrite this sentence
Line 129 Of the 91 selected ILs in the background of WGL14(RP2),
Line 131 delete “and”.